# OpenReview forum: "CoopEval: Benchmarking Cooperation-Sustaining Mechanisms and LLM Agents in Social Dilemmas"
_ICML.cc/2026/Conference — ICML 2026 regular_

### Official Review · Reviewer_vj16 · 2026-02-24

**Soundness:** 2
**Presentation:** 2
**Significance:** 4
**Originality:** 3
**Overall Recommendation:** 4
**Confidence:** 4

**Summary:**

The paper aims to create a benchmark of previously studied cooperation mechanisms in societies of LLM agents. It studies four mechanisms of cooperation: repetition (direct reciprocity); reputations (indirect reciprocity), contracts and delegation, across a variety of social dilemmas. The paper finds that some mechanisms are substantially more capable of promoting cooperation than others.

**Compliance With Llm Reviewing Policy:**

Affirmed.

**Final Justification:**

I am keeping my final score (Weak Accept) after the rebuttal with the authors. This score reflects the significance of the paper, which points in a genuinely important direction as many methods to improve cooperation in LLMs have been proposed but never equally compared, but has flaws in presentation and in clarifying the implementation of the methods. Nevertheless, I believe it is a valuable contribution, provided the authors implement the proposed changes which addressed some of my issues.

**Key Questions For Authors:**

Q1: The paper is labelled as relying on “morality-agnostic” mechanisms. However, is the application of a reputation system, which is dependent on social norms to dictate which behaviors are acceptable, not a formalization of a moral system?

Q2: Given that LLMs are themselves still largely delegated by humans, is there not a problem of creating a lack of transparency to users by further delegating their requests via your mediation mechanism, as well as raised problems of accountability? Is this an adequate method to use in real world settings?

Q3: As noted by the introduction, some LLMs when not prompted to be selfish will typically be more prosocial. My question is why aim to develop mechanisms to improve cooperation in LLMs while simultaneously providing them with instructions to be selfish and maximize their own payoffs. I understand that your goal is to study if LLMs will be then influenced by the mechanism to cooperate regardless. Yet, this makes it not reflect current LLM systems under natural circunstances. Could you elaborate on why this choice was made, and its potential implications to the results?

Q4: I would like to better understand the reasoning behind the contract mechanism. It appears that by giving the LLM the decision to formulate a contract via language, you are already giving it much more expressivity than any other mechanism. As such, its performance increase over, say, reputations, is not entirely surprising. A fairer comparison would be if LLMs could also discuss about what social norms they use, or would gossip about each other’s reputations, or if it had to pick between a set of contracts. Do you have insights regarding if other ways to create contracts would lead to performance impacts?

**Limitations:**

The paper should be more clear about the limitations of the methodology. In particular, it should explain other ways to implement these cooperation mechanisms, what deviations it made for the sake of translating these mechanisms to LLM settings, and what are the consequences of such deviations. Furthermore, the social implications and resilience of these mechanisms, in the context of both LLM-LLM but also human-LLM interactions should be discussed. In human-LLM scenarios, the "implementations" of these mechanisms might not match those of humans, and the negative consequences of this mismatch should be discussed.

**Strengths And Weaknesses:**

General appreciation

The paper offers an important overview of how cooperation mechanisms typically used by humans or in multi-agent settings fare in societies of LLMs. This is particularly relevant, given that it allows us to compare mechanisms that have previously been studied in isolation under identical implementation settings. The social dilemmas studied, and the mechanisms used, are adequate and representative of those most studied in game theory and multi-agent settings. Nevertheless, there are some questions regarding the implementation of some of these methods that appear to deviate from their traditional implementations.

Major issues

From what I understood, the given definition of reputation is based on sharing the history of interactions. However, this does not match a large portion of studies in indirect reciprocity, which are based on sharing past reputations. In other works, a social norm would be defined that assigns a good or bad reputation given both the action done by an individual and the reputation of the individual the action was done to (and may even include the reputation of the individual doing the action). Cooperation under indirect reciprocity relies that this social norms is common so every agent assigns reputations the same way. This is the implementation also followed by many works on LLMs and MARL under reputation systems (see the work of Smit, Martin, and Fernando P. Santos. "Learning fair cooperation in mixed-motive games with indirect reciprocity.", or Ren, Tianyu, and Xiao-Jun Zeng. "Reputation-based interaction promotes cooperation with reinforcement learning."). As such, despite the work aiming to do a comparison between cooperation mechanisms, I do not think it is currently following the literature when implementing indirect reciprocity.

Figure 1 requires a clearer caption. Since it presents figures with humans, it suggests that the work presented is somehow applied to humans. Furthermore, some sections of the figure are unclear: In the repetition panel the two players appear after the first two games, making it seem like they have not played the first games. The reputation system does not make clear whose reputation is being affected or if it is a general global reputation. And the mediation panel has no title and it is unclear what happens in the thought bubble.

According to the guidelines, the abstract must be a single paragraph, as opposed to two as it currently is.

The LLMs are tested using a temperature to 1 and are asked for a probability density function over actions instead of prompted to provide a decision. However, not only does this hurt reproducibility efforts, but also may significantly impact the results compared to asking them to select an action (despite the reference given regarding random choices). Additional studies should be conducted to include these variations in order to confirm their impact and compare their results with previous studies.

Minor issues

In line 207 right column, while clear for readers with a background in game theory, it would be useful to explain (either in the main text or an appendix) why it is relevant to not know when the game ends.

Line 229 left column mentions that there is no consensus on social norm complexity. Work such as Santos et. al’s “Social norm complexity and past reputations in the evolution of cooperation” offer insight on this and shows a typical implementation of social norms.

Some choice of words should be made more formal. In particular, referring to LLMs as “nice” or “nicer” is unclear and suggests dimensions not expressed by your results.

Other issues with the paper are expressed in the questions section, as they require clarifications.

---

> ### Author Rebuttal · Authors · 2026-03-31
>
> We thank the reviewer for their valuable feedback, we will incorporate the "minor issues" and the great discussion points from below in the final version of the paper.
> # W1, W6, Q1:
> It is correct that we do not encode social norms directly into our Reputation mechanisms, and that there is extensive prior work on social-norm-based reputation systems (some of which we already cite in Appendix A). We did not experiment with those because they typically assume the population has already acquired or learned a social norm, under which the questions then become “Are agents incentivized to follow the social norm? If everyone follows the social norm, will the outcome be cooperative?” Since the social norm defines what characterizes “good” and “bad” actions, it is also essential to establish notions of morality in this mechanism (Alexander, 1987, The Biology of Moral Systems, Ch. 2.1).
>
> In contrast, our work investigates the other popular mechanism variant in the literature in which every player is presented with a history of the past and assesses their co-players independently (see Vallinder and Hughes, 2025; Okada, 2020; and the references therein). In this bottom-up approach, social norms may _emerge_ and evolve in this multi-agent system in a decentralized fashion (as opposed to a top-down pre-defined norm). We believe it makes the mechanism more comparable to the Repetition mechanism, and in this paper we generally pursue solutions arising endogenously.
>
> We will add a paragraph on social-norm-based reputation mechanisms to the main body, and that it forms an important direction for future work.
> # Q2
> We share the concerns for mediators whose inner workings we do not understand or cannot audit (e.g., LLM mediators or Google maps routing). Our implementation, however, works with rule-based open mediators: each player is aware of what the mediator would do in each scenario. (In complex real-world settings, where enumerating scenarios becomes infeasible, the mediator might instead be, say, a well-documented open-source code repo, accessible to everyone.) This makes the mediator transparent and auditable to the end user, and keeps the LLM player accountable for delegating if and only if it is sensible to do so on behalf of the end user.
> # Q3
> Great question. We only ask the LLM to “to maximize [its own] total points received in the game”. In game theory, we assume the utility function reflects _everything_ a player cares for. If a player cares about the outcome of another player, this needs to be incorporated in the utility functions. (Say, a company telling the LLM to care for its return on investment, but also that this does not cause great suffering for others, plus other criteria.) As an aligned and rational agent, the LLM should then _only_ optimize for this specified utility function, hence our prompt. If, instead, we tried to pursue cooperation simply by making every player care about the sum of "utilities", then the agent’s real utility would be that sum, and we would have simply assumed away the problem. Our view is that there will likely be agents interacting in the world with different utility functions, and we want to study mechanisms that enable cooperative outcomes, even if players optimizing for their own representative utility function would otherwise lead to a game-theoretic dilemma.
> # Q4
> Since we don’t let the agents communicate in advance in any of the mechanisms, it would not be fair to support that in Reputation. In an implementation of a social-norm-based reputation mechanism, a better analogue could be to let every player propose a social norm and vote, and to track the reputation of every player using the winner norm. This fits greatly to our call “for future work when it comes to understanding and progressing indirect reciprocity in LLMs”.
>
> Moreover, we recall that Contracting has multiple caveats. It is theoretically more powerful than the other three mechanisms and not always viable (Sec. 3). We also need to be careful in its design in order to actually fit Theorem 1, because other previously explored designs can (and ought to) be used to _exploit_ other agents (l.264 onwards). We view more foundational work on discovering other viable designs as important.
> # W4
> As acknowledged, LLMs struggle to faithfully implement mixed strategies when forced to select a single action repeatedly. Since our focus is on game-theoretic reasoning, and not on accurately implementing strategic distributions through logits, we set things up to investigate the former only. But if the reviewers believe for it to be a valuable contribution, we are happy to add experiments on this.
> # More Evals!
> Further analysis only described to other Reviewers:
> - Analysis of LLM’s chain-of-thought (CoT) reasoning justifications
> - Frequency of exploitative/forgiving/initially nice behavior in Repetition/Reputation
> - and Ablations on Continuation Probability and History Window Size
> - Design Quality and Voting in Mediation Contracting

---

> > ### Author Rebuttal · Reviewer_vj16 · 2026-04-01
> >
> > Some of my major concerns, such as the temperature used to query the LLM, were not addressed. Furthermore, while the usage of bottom-up norms has been used before, its efficacy compared to top down norms is substantially different, and its inclusion may be misleading to readers, specially in this context where methods are being compared. While the questions were partially answered, but some relevant aspects, such as Q3, seem very relevant for the outcome of the methods. As I feel this won’t be solved by a rebuttal, I will keep the current score.

---

> > > ### Author Response · Authors · 2026-04-07
> > >
> > > We thank the reviewer for their additional comments.
> > >
> > > # Re: LLM temperature
> > > We responded to the major concern that included this point in our paragraph `W4`, and are not sure whether the reviewer was not satisfied with the response, or whether the reviewer missed it (which is quite possible because of our concise, but non-descriptive paragraph titles).
> > >
> > > We are also not sure if we understood their concern revolving around LLM temperature:
> > > - Would they have liked us to run our experiments with a temperature of 0 instead of 1? We find the fact that LLMs are generative models (i.e., not to always return the same response tokens) an important aspect to consider when it comes to the practical deployment and safety of LLMs for decision making, which is why we included the nondeterminism in our setup and counterbalanced it with repeated experiments / multiple decision samples. This experiment design allows us, for example, to observe that in the repeated prisoner’s dilemma and under an LLM temperature of 1, GPT-4o cooperates in round 1 only 72.2% of the time (i.e., when there was no history available yet). In particular, even though cooperating is its most likely action, there is still a non-negligible likelihood that GPT-4o will enter game dynamics that are quite difficult to recover from! Qwen-30b’s numbers are even lower (at 44.4%); so low, in fact, that Qwen-30b’s most likely action in that situation is to defect.
> > > - If, on the other hand, the reviewers wish that we run ablations on how much the LLM temperature affects our results, we are happy to do that in our revised version of the paper.
> > >
> > > # Re: Reputation
> > > We do not intend to mislead the reader about the bottom-up design of our Reputation mechanisms. If the reviewer has any suggestions on how to make that more clear, we would be happy to hear about it. As mentioned earlier, our revised version of the paper will include in the main body a discussion on this approach versus the social norm approach and emphasize that the latter as an important direction for future work. Additionally, we will add a reminder in the results section about our design choice  (more precisely, when we discuss our results for the reputation mechanisms).

---

### Official Review · Reviewer_QP65 · 2026-03-12

**Soundness:** 3
**Presentation:** 3
**Significance:** 2
**Originality:** 2
**Overall Recommendation:** 4
**Confidence:** 4

**Summary:**

This paper presents an experimental study where LLM agents interact in social dilemma games, under 4 different coordination mechanisms. The key point, as I understand, is not the proposal of coordination algorithms, but just to show how current LLMs behave when playing the games with no intervention, and how that behaviour changes when there is a mechanism in place. Results show that modern LLMs tend to play in a non-cooperative manner, but some of the mechanisms are able to effectively increase the social welfare.

**Compliance With Llm Reviewing Policy:**

Affirmed.

**Final Justification:**

The work is interesting overall, but the evaluation is only performed on small problems.

There was an interesting discussion in the rebuttal regarding sequential social dilemmas. Overall I think it is ok to study non-sequential social dilemmas, of course, it is just that the evaluation could then be completed with more complex problems afterwards.

That being said, even if we accept sequential social dilemmas for future work, the number of agents and actions studied was quite small.

Hence, I kept my original score, but that was already tending towards acceptance (Weak Accept).

**Key Questions For Authors:**

1 - Do I understand correctly that the key contribution is the experimental study, and there are no significant technical innovations?

2 - The mediators and contracts created by the LLM agents are reasonable?

3 - Do I understand correctly that the result in Theorem 1 is not new?

**Limitations:**

Limitations and ethical considerations seem well discussed in Future Research and Impact Statement.

**Strengths And Weaknesses:**

Strengths:

  - The paper is interesting, overall. It is interesting to see how LLM agents may behave in isolation, and how that can be changed using classical cooperation mechanisms.

  - The paper is well situated in the literature, including the fundamental work done before LLM agents.

  - The overall experimental framework is well designed.

  - There is one theoretical result regarding the guarantees of the mechanisms to reach desirable outcomes. However, I think that was already known in the literature, and the point was to show all of them under the same theoretical framework, and in the context of the paper.

Weaknesses:

  - The paper does not really present any new algorithm or mechanism, the key point is the experimental setup and the results.

  - An experimental study on social dilemmas is also not new by itself, for sure; so their contribution is limited to studying these ideas in the context of LLMs.

  - The evaluation was done in very simple abstract games. It would be interesting to see results in more complex games (e.g., Cleanup, Harvest). Even in single-shot games, large action spaces could also be studied.

  - Impact of parameters (e.g., \delta, k) was not studied.

Detailed Comments:

- Table 1, Trust game could be more clear? Also, when you say "P2 cannot observe P1’s decision, but regardless, P1’s business multiplies the total investments by a factor of 4.", I think you meant "P2's business", right? It is the amount of money that P2 receives that gets multiplied by 4, and that may or may not be divided across the 2 players.

- "we eliminate P2’s strategy to share the returns since that one is strictly dominated.." -> Double "."

- "We follow the standard approach of deciding whether an another round is played via a biased coin flip after each round." -> I think just "whether another round" or "whether one more round is played"

- Paragraphs are missing indentation.

- "We also include plots of all match up payoffs for each base game and mechanism Section F" -> "in Section F".

- "We see stark differences in the effectiveness of our cooperation mechanisms." -> Well, I think the paper is about studying existing mechanisms, and not proposing new ones? So to avoid misunderstanding should be something like: "We see stark differences in the effectiveness of the cooperation mechanisms that we study".

- "and that a few non-cooperative actors could suffice to poison the well for everyone’s interactions" -> "poison the welfare"?

- "Another open direction is to investigate the robustness of our cooperation with regards to more purposefully built LLM agents" -> Again, I don't think these are "your" mechanisms, it should be something like "the robustness of the cooperation mechanisms studied with regards to (...)"

---

> ### Author Rebuttal · Authors · 2026-03-31
>
> We thank the reviewer for their detailed feedback and comments, which we have incorporated.
> # Q1, Q3, W1, W2
> Correct, we rely on established mechanisms from the literature. Note that Mediation for LLMs has not been implemented before. The approval voting component to Mediation/Contracting is (to our knowledge) also conceptually novel. The theory on these mechanisms usually
> - assumes a good mediator is given, while sidestepping how the it is being decided on, or
> - grants the power to design a contract to one agent. This leads to unfair welfare distributions, and cannot sustain cooperative outcomes (Sec. 3).
>
> So we needed to adjust the Thm 1 proofs to this new voting component.
>
> Our _main_ technical contribution is to put cooperation mechanisms for social dilemmas under one consistent umbrella, in terms of formal treatment and experimental framework. While prior work compared at most two mechanisms (e.g., repetition and reputation in studies on human subjects) on one social dilemma, our paper offers a scalable foundation for incorporating a plethora of cooperation mechanisms. It serves as a valuable guide to mechanism/environment designers and LLM agent designers who seek to elicit cooperative behavior.
> # W3
> Our social dilemmas already cover many actions, many players, and asymmetry.
>
> As written in conclusion, extending to sequential social dilemmas (Cleanup, Harvest, etc.) is an interesting direction for future work. At the same time, because these games are iterative, they inherently embed a Repetition component. Our goal in this paper is to study the effects of different mechanisms _in isolation_.
> # Q2: How good are the mediators / contracts?
> Voting and Acceptance:
>
> Due to the voting process, one well-designed mediator/contract often suffices in order to establish cooperation amongst the LLM models, especially under contracting.
> - Unanimous approval occurs 70%-90% of the time (with exceptions: Mediation x PublicGood and Contract x Trust), showing mutual support from players to the LLM designed mediators/contracts.
> - The winning contract proposal is accepted by all players at even higher rates, and the action decision thereafter appears the most straightforward in terms of CoT reasoning complexity
> - GPT-4o & Qwen-30b struggle to delegate to the winning mediator proposal. This explains:
>     - lower Mediation welfare when these models participate, as they sometimes do not delegate even when they should. (whereas in Contracting, the stronger models under an active contract will still secure a high utility)
>     - why Mediation recovers comparable performance to Contracting after evolutionary pressures have been applied
>
> Designing Phase (pre-voting):
>
> Additionally, our analysis confirms high variance in whether proposed mediators/contracts are game-theoretically sound, depending heavily on the social dilemma and LLM model.
>
> For a proposed contract, we test how often everyone playing the (most) cooperative action forms a Nash equilibrium (NE) or is weakly dominant (WD) if the contract payoff changes went into effect. We do similarly for a proposed mediator, except that we have to check that for “everyone delegating”, and that the mediator then indeed plays the (most) cooperative action for everyone.
>
> Mediators:
> - In PrisonersDilemma (PrisD) and PublicGoods (PublG), NE and WD are satisfied at the same rates. In TravellersDilemma (TravD) and TrustGame (TrustG), WD is theoretically not achievable.
> - NE is satisfied most often in TrustG (89% of the time), PrisD (81%), PublG (69%, GPT-5.2 only 33%), and then TravD (64%, Qwen-30b & GPT-4o fail).
>
> Contracts:
> - Very high rates for WD across LLM models in PublG (overall, 94%) and PrisD (81%, Qwen-30b fails), significantly higher than NE rates in Mediation
> - TrustG (NE and WD) is hard for every model (50%-67%) except Claude
> - Many models fail TravD, including Claude for WD
>
> # W4: Ablations
> We performed ablations of Repetition and Reputation on PrisonersDilemma (defaults continuation probability d=0.8, history window k=3), varying either [d in 0.7, 0.9] or [k in 2, 4]. Summarized findings:
> - Repetition. Highly insensitive to d and k, and slightly higher Mean (making it competitive with Mediation now). Replicator dynamics still converges to cooperative mean of ~2.0. This replicates similar LLM insensitivity results for Iterated PrisD to window sizes (Fontana et al., 2025, Figure A8) and conti. probs. (Pal et al. 2026, arXiv:2601.09849, p.6).
> - Reputation. Shows an inverse effect: Lower d or k correlates with a higher Mean (between 1.32 to 1.54). For d, this is counterintuitive, as lower continuation probability should make agents care less about future rounds, and thus cooperate less. Reputation+ is now competitive with Reputation-.
> # More Evals!
> Further analysis only described to Reviewer AAWS:
> - Analysis of LLM’s chain-of-thought (CoT) reasoning justifications, via LLM as a judge
> - Frequency of exploitative/forgiving/initially nice behavior in Repetition/Reputation

---

> > ### Author Rebuttal · Reviewer_QP65 · 2026-04-04
> >
> > Thank you for the clarifications.
> >
> > I think the paper is interesting, but I still think the evaluation is somewhat restricted to toy problems. While sequential social dilemmas have "repetition", they still lead to non-cooperative behaviour, and perhaps identity of agents could be hard to keep track in large groups.
> >
> > Similarly, the number of agents and actions is very small on the problems studied (at most 3 agents and at most 4 actions), so I don't think my point W3 was addressed.
> >
> > Hence, I think I will keep my current score.

---

> > > ### Author Response · Authors · 2026-04-07
> > >
> > > We are happy to see that the reviewer seems satisfied with the additional evaluations and analysis we ran, and thank the reviewer for the follow-up question on W3, since our previous response ran into space constraints.
> > >
> > > We agree with the reviewer that large-scale sequential social dilemmas are an interesting direction for future work. At the same time, we believe that for our research questions it makes sense to study traditional social dilemmas from game theory, such as the four dilemmas we implemented:
> > > - The advantage of traditional social dilemmas for our experiments was that they are simple enough to allow us to study the effects of cooperation mechanisms and game components (e.g., actions, players, asymmetry) in an isolated fashion
> > > - Sequential social dilemmas (such as Cleanup, Harvest, etc.) are richer, multifaceted “toy problems” and interesting precisely because they bring multiple mechanisms and game aspects together (e.g., iterative play, which embeds a Repetition mechanism component; or the difficulty of tracking agent identities in large groups, which is also a concern in the Reputation mechanisms we study).
> > >
> > > In light of our paper’s goal to provide a comparison of individual cooperation mechanisms studied in the literature, does the reviewer agree with our assessment?

---

### Official Review · Reviewer_AAWS · 2026-03-20

**Soundness:** 2
**Presentation:** 4
**Significance:** 3
**Originality:** 3
**Overall Recommendation:** 5
**Confidence:** 4

**Summary:**

The paper studies the cooperative behavior of LLM agents in social dilemmas under different "cooperation mechanisms". The social dilemma games (prisoner's dilemma, traveller's dilemma, the public goods game, and the trust game) focus in on the  conflict between individual gain and collective welfare while the four mechanisms studied (repetition, reputation, mediation, and contracting) all provide pathways toward cooperation being sustained in equilibrium. Across all game$\times$mechanism pairs and 6 models, the authors show that, in the absence of a cooperation mechanism, modern LLMs tend to defect rather than cooperate. They also show surprisingly dramatic differences in efficacy between the different mechanisms with reputation resulting in the most modest improvement in outcomes and contracting moving behavior closest to social optima. The benefits of the cooperation mechanisms is increased when the representation of different models in the population undergo replicator dynamics. The authors also show differences between the cooperative behavior of the different models with Gemini models coming out ahead.

**Compliance With Llm Reviewing Policy:**

Affirmed.

**Final Justification:**

The rebuttal partially addressed my concerns, increasing my score from a 4 to a 5. I think the work would still greatly benefit from a more careful dissection of model behavior.

**Key Questions For Authors:**

1. See questions in weakness 1 above.
2. At temperature 1.0, is 3 trials per instance sufficient? Could you instead obtain the logprobs directly (and potentially sample from this distribution yourself like y) to get a more complete picture without having to run up experiment costs?
3. Regarding the Promising Trends paragraph of Section 6, is it really all that surprising that the replicator dynamics boosted cooperation given that they select for better performing models and, in social dilemmas, this aligns with cooperative behavior? Is there something I am missing that makes this counterintuitive?

**Limitations:**

yes

**Strengths And Weaknesses:**

## Strengths
1. **Comprehensive empirical work on a timely subject**

The authors' experimental setup covers a large landscape with a multitude of further open problems, making it fruitful ground for future research. Their codebase also appears to lend itself toward extensibility, and I could foresee it growing and/or being built upon by others.

2. **Two-sided audience and applicability**

The work provides a useful benchmark for the training of more cooperative LLMs as well as important guidance on mechanism/environment design for those seeking to elicit cooperative behavior.

3. **Cross-mechanism and cross-model investigation**

The paper includes direct comparisons of the efficacy of different mechanisms and the study of heterogenous cross-play with population dynamics. Both of these settings are naturally of interest and relatively understudied with many works studying a single mechanism or homogenous or static agent populations.

4. **Surprising findings on mechanism efficacy**

The low efficacy of reputation mechanisms and the high efficacy of contracting are not a priori expected. This is a practically useful finding and also opens up many questions for further inquiry.
## Weaknesses
1. **Explanatory depth**

While the findings on different levels of cooperation are interesting, at present I feel they tell only a partial story and, at times, leave more questions than answers. When agents fail to cooperate, what breaks down? Is it a comprehension problem (e.g., do they struggle to make sense of co-player histories)? Is it a game-theoretic reasoning problem where they struggle to understand best response dynamics? Similarly, when they succeed, what enables them to find cooperative equilibria? Are mediation and contracting more successful because they require less dynamic reasoning on the part of the agent? It is understandable that the authors cannot answer all questions in one shot, but some greater efforts or discussion around these questions is important. I think these questions are most salient and most abundant for the two extreme performers (contracting and reputation) so I elaborate on these below.

**Contracting**: The quality of proposed contracts and the voting behavior of the agents are both essential determining factors for the outcome, yet they receive little to no attention in the current treatment. The contract, and thus outcome, also varies each round based on the proposals and voting so I think more clarity is needed on what it means to say that contracting is an effective mechanism for cooperation. For example, how much does its efficacy vary based on the participating models? If all the models participating are relatively weak, does the efficacy of contracting also suffer significantly due to the lack of a "leader" to put forth a good proposal?

**Reputation**: How well do agents infer information from the history? Are they updating beliefs about their co-players and, if so, in what manner? Is the key breakdown in this inference from the history or is it in choosing a strategy from a more complex space (length-k histories $\rightarrow\Delta(A_i)$)?

2. **Lack of ablation across key parameters**

The authors fix the values of several parameters throughout including continuation probability (0.8) and history length (3). For continuation probability, the selection is justified as being sufficient for cooperation under Theorem 1. Theorem 1; however, assumes rational agents. Given that LLMs do not fit this category, there is little reason to expect the theoretical sufficiency of $\delta=0.8$ to carry over. The same justification (sufficiency in Theorem 1) is given for selecting $k=3$ and it raises the same concern. Lacking a strong theoretical basis for a fixed value, I think ablations are important, even if they are only performed on some subset (say for the best performing model in other experiments or for one of the four games).

---

> ### Author Rebuttal · Authors · 2026-03-31
>
> We thank the reviewer for their detailed feedback and comments, which we have incorporated.
> # Q1, W1: Evals on CoT Justifications, and Repetition / Reputation
> To analyze reasoning, an LLM-as-a-judge evaluated CoT responses against 15 justification categories (non-exclusive, and provided to the judge as title-description pairs, referenced in quotes below).
>
> In Repetition and Reputation, we analyzed each LLM model’s rate of cooperation conditional on co-player’s last round action(s), to get an approximate understanding of whether they tend to (respectively) exploit, forgive, and/or be initially nice (i.e., cooperate in the first round). The reviewer also asked about the LLM’s beliefs; it is not clear to what elicitation method is best for that?
>
> Summary of the two additional evaluations.
>
> General:
> - Aside from a few Gemini-R responses, no LLM responses under _NoMechanism_ include any arguments that could possibly justify cooperating, such as “Social Welfare”, “Trust”, etc.
> - For every mechanism and model, the most commonly present justifications are “Individual Utility Maximization” and “Strategic Equilibrium Focus” (except the latter for the old model GPT-4o). This demonstrates some extent of LLM understanding that when the mechanisms are in place, even selfish agents might be best off cooperating
> - GPT-5.2 is the least concerned with considerations involving “Strategic Influence over the future dynamics of the game”, “Uncertainty about Other Players”, and (after GPT-4o) “Strategic Equilibrium Focus”, indicating a decision making disadvantage in terms of multi-agent and long-term thinking
> - We had described examples where GPT-4o would not pick a dominant action deterministically in order to “stay unpredictable”. Now we know concretely that many of GPT-4o’s decisions are based on “Uncertainty about Other Players” (\~65%) and/or “Exploration-Exploitation Trade-off” (\~15%, which is >4x higher than for the second highest model)
>
> Focusing on Reputation and Repetition:
> - The Reputation mechanisms show the highest rates in “Uncertainty about the other players' intentions or strategies” (at 58% frequency)
> - Only Repetition x {Gemini, Claude} show occasional considerations of “Reciprocity”
> - These repetitive mechanisms are also the only ones where “Strategic Influence” or “Trust” frequently play a role in the LLM decisions
>
> - In the first round of Reputation (No History), there is a hesitancy to cooperate in the trust game across the LLM models, and a staggering 50%-100% rate of free-riding and undercutting in the public goods game and trust game (excluding the Gemini models).
> - Surprisingly, LLM models in the reputation mechanisms are _less cooperative_ towards last round cooperators than towards agents lacking history
> - The two previous points, together with 80%-100% defection rates against last round defectors, explains why we barely see cooperation (sustained across rounds) in the reputation mechanisms
> - The social dilemmas PublicGood and TrustGame seem to be generally challenging to GPT-5.2, GPT-4o, and Qwen-30b, since those models exhibit high defection rates even under Repetition. This speaks to an environment _generalization gap_ of Axelrod’s (1984) cooperation principle “never [be] the first to defect” for the PrisonersDilemma
> # Q2
> We repeat each individual experiment 3 times due to combinatorial scale (Mechanism x Game x LLM_P1 x LLM_P2 x …). This sums to 8586 decisions per LLM model, or >50k in total. While we agree that there is not enough signal to deduce performances in an individual experiment, we instead describe our results in terms of, and obtain strong signals from, aggregated experiments.
>
> We had considered the idea of obtaining logprobs directly, but found it difficult to apply to our response format that requires the use of multiple tokens—for defining the action probability distribution (cf. ll.293-297 right) and for allowing chain-of-thought reasoning.
> # Q3:
> The results are not obvious nor given. In the single-shot prisoner’s dilemma, evolutionary dynamics will always converge to the LLM model that defects the most, because defecting is strictly dominant. When one of the cooperation mechanisms we study is in place, there will be a plethora of equilibria which the replicator dynamics could converge to. Which one it will be highly depends on the strategies we observe in the LLM models. Always defecting in the iterated prisoner’s dilemma (IPD) is one such equilibrium, and the population would converge to it if our experiment had shown that many LLMs cooperate unconditionally or defect most of the time. Fortunately, enough of them seem to understand the upsides of cooperative _equilibrium_ strategies, such as Tit-for-Tat in IPD.
> # More Evals!
> Further analysis only described to Reviewer QP65:
> - Ablations on continuation probability and history window size in Repetition / Reputation
> - Proposed mediators / contracts regarding their quality, the voting, and delegation/acceptance rates

---

> > ### Author Rebuttal · Reviewer_AAWS · 2026-04-04
> >
> > I thank the authors for their response. While I maintain that the work would greatly benefit from further explanatory analysis of the findings from evaluation, the authors' rebuttal did partially address my concern. I will bump my score to a 5.

---

### Decision · Program_Chairs · 2026-04-30

**Decision:**

Accept (regular)

**Comment:**

This paper investigates the cooperative behaviour of LLM agents in social dilemmas under four distinct cooperation problems: prisoner's dilemma, traveller's dilemma, public goods game, and trust game. Each of these represents the tension between individual gain and collective welfare. In addition, the mechanisms examined in this study (repetition, reputation, mediation, and contracting) represent principled pathways through which cooperation can be sustained in equilibrium. Across all game-mechanism pairings and six models, the findings reveal that, without any cooperation mechanism, modern LLMs default to defection. Strikingly, the mechanisms vary dramatically in efficacy: Reputation yields only modest gains, whereas contracting drives behaviour closest to the social optimum. These benefits are further amplified when model representation in the population evolves under replicator dynamics. The study also uncovers meaningful differences across model families, with Gemini models exhibiting the strongest cooperative tendencies overall.

The overall consensus is that while the experimental results can be further improved, the novelty and findings of this paper is worth of being presented at the conference. Hence I recommend acceptance.